# Left atrial reservoir strain is a marker of atrial fibrotic remodeling in patients undergoing cardiovascular surgery: Analysis of gene expression

Toshiaki Nakajima[1][¤]*, Akiko Haruyama[1], Taira Fukuda[2], Kentaro Minami[1], Suguru Hirose[1], Hiroko Yazawa[1], Takafumi Nakajima[1], Takaaki Hasegawa[1], Yoshiyuki Kitagawa[1], Syotaro Obi[1], Shu Inami[1], Gaku Oguri[3], Ikuko Shibasaki[4], Hirohisa Amano[1], Takuo Arikawa[1], Masashi Sakuma[1], Shichiro Abe[1], Hirotsugu Fukuda[4], Shigeru Toyoda[1]

1 Department of Cardiovascular Medicine, Dokkyo Medical University and Heart Center, Dokkyo Medical University Hospital, Mibu, Tochigi, Japan, 2 Department of Liberal Arts and Human Development, Kanagawa University of Human Services, Kanagawa, Japan, 3 Department of Cardiovascular Medicine, University of Tokyo, Tokyo, Japan, 4 Department of Cardiovascular Surgery, Dokkyo Medical University Hospital, Dokkyo Medical University, Mibu, Tochigi, Japan

¤ Current Address: Heart Center, Dokkyo Medical University, Mibu, Tochigi, Japan
* nakat@dokkyomed.ac.jp

**Data Availability Statement:** All relevant data are within the manuscript and its Supporting Information files.

## Abstract

Left atrial strain (LAS) measured by two-dimensional speckle tracking echocardiography (2DSTE) is considered to be a marker of LA structural remodeling, but it remains unsettled. We investigated the potential usefulness and clinical relevance of LAS to detect atrial remodeling including fibrosis by analyzing gene expression in cardiovascular surgery patients. Preoperative 2DSTE was performed in 131 patients (92 patients with sinus rhythm [SR] patients including paroxysmal AF [PAF], 39 atrial fibrillation [AF]) undergoing cardio-vascular surgery. Atrial samples were obtained from the left atrial appendages, and mRNA expression level was analyzed by real-time reverse transcription polymerase chain reaction (RT-PCR) in 59 cases (24 PAF, 35 AF). Mean value of left atrial reservoir strain (mLASr) correlated with left atrial volume index (LAVI), and left atrial conduit strain (mLAScd). mLASr also correlated with left atrial contractile strain (mLASct) in SR patients including PAF. mLASr was significantly lower, and LAVI was higher, in the AF group, compared with SR patients including PAF. The expression of COL1A1 mRNA encoding collagen type I α1 significantly increased in AF patients (p = 0.031). mLASr negatively correlated with COL1A1 expression level, and multivariate regression analysis showed that mLASr was an independent predictor of atrial COL1A1 expression level, even after adjusting for age, sex, and BMI. But, neither mLAScd / mLASct nor LAVI (bp) correlated with COL1A1 gene expression. The expression level of COL1A1 mRNA strongly correlated with ECM-related genes (COL3A1, FN1). It also correlated ECM degradation-related genes (MMP2, TIMP1, and TIMP2), pro-fibrogenic cytokines (TGFB1 encoding TGFβ1, END1, PDGFD, CTGF), oxidant stress-related genes (NOX2, NOX4), ACE, inflammation-related genes (NLRP, IL1B, MCP-1), and apoptosis (BAX). Among the fibrosis-related genes examined, univariable regression

**Funding:** This study was supported in part by JSPS KAKENHI Grant Number 19H03981 (to T.N.), 22H03457 (T.N.) and the Vehicle Racing Commemorative Foundation (to T.N.).

**Competing interests:** The authors have declared that no competing interests exist.

**Abbreviations:** LA, left atrium; LV, left ventricle; AF, atrial fibrillation; PAF, paroxysmal atrial fibrillation; SR, sinus rhythm; LAS, left atrial strain; LASr, LA reservoir strain; LAScd, LA conduit strain; LASct, LA contractile strain; 2DSTE, Two-dimensional speckle tracking echocardiography; LVEDP, left ventricular end-diastolic pressure; LAD, left atrial diameter; LVEF, left ventricular ejection fraction; LAVI, left atrial volume index; PDGF, platelet-derived growth factor; CTGF, connective tissue growth factor; ROS, reactive oxygen species; TGFβ, transforming growth factor beta; NOX, NADPH oxidase; ECM, extracellular matrix; ANP, atrial natriuretic peptide; MR, mitral regurgitation; MVR, mitral valve replacement; MVP, mitral valve plasty; AS, atrial stenosis; AVR, aortic valve replacement; CABG, coronary artery bypass grafting; AR, aortic regurgitation; BMI, body mass index; DE-CMR, delayed gadolinium-enhanced cardiac magnetic resonance; BAX, Bcl-2-associated X protein; MMP, matrix metallopeptidase; TIMP, tissue inhibitor of metalloprotease; ROC, receiver operating characteristic curve; RT-PCR, Real-time quantitative reverse transcriptase/polymerase chain reaction.

analysis showed that log (COL1A1) was associated with log (TGFB1) (adjusted $R^2 = 0.685$, p<0.001), log (NOX4) (adjusted $R^2 = 0.622$, p<0.001), log (NOX2) (adjusted $R^2 = 0.611$, p<0.001), suggesting that TGFB1 and NOX4 was the potent independent determinants of COL1A1 expression level. mLASr negatively correlated with the ECM-related genes, and fibrosis-related gene expression level including TGFB1, NOX2, and NLRP3 in PAF patients. PAF patients with low mLASr had higher expression of the fibrosis-related gene expression, compared with those with high mLASr. These results suggest that LASr correlates with atrial COL1A1 gene expression associated with fibrosis-related gene expression. Patients with low LASr exhibit increased atrial fibrosis-related gene expression, even those with PAF, highlighting the utility of LAS as a marker for LA fibrosis in cardiovascular surgery patients.

## Introduction

The left atrium (LA) contributes to cardiac hemodynamics by modulating left ventricular (LV) filling via reservoir, conduit and contractile functions [1]. Patients with cardiovascular diseases, such as valvular heart disease, undergo LA remodeling and alterations in LA function, leading to atrial fibrillation (AF) [2]. LA remodeling is associated with increases in LA structure remodeling including fibrosis, an important prognostic marker and clinical outcome [3, 4]. Two-dimensional speckle tracking echocardiography (2DSTE) is feasible for assessment of myocardial LA function and deformation [5, 6]. In particular, LA reservoir strain (LASr) is a positive strain that represents LA stretching during LV systole as an index of LA reservoir function [5, 6]. It is influenced by LA stretch due to pressure or volume load and LA relaxation/stiffness, and it is secondary to increased LV filling pressure, left ventricular end-diastolic pressure (LVEDP), and LV diastolic function [7–9]. Thus, quantification of LASr can provide insights into LA mechanics in several pathophysiological conditions, including those in patients undergoing cardiovascular surgery.

Deposition of extracellular matrix (ECM) leads to fibrosis, which plays essential roles in structural remodeling and functional changes in AF [10, 11]. The major components of the ECM are collagen type I, collagen type III, and fibronectin [12]. Among them, a heterodimer of the α-subunits of collagen type I is the collagenous product of cardiac fibroblasts, accounting for 80% of the total content [13]. The cellular and molecular control of atrial fibrosis is highly complex, but the process of atrial fibrosis includes various individual and multifactorial processes with underlying complicated interactions between cellular and neurohormonal mediators [14]. Several profibrotic signaling pathways [2, 15, 16], such as the renin/angiotensin system, transforming growth factor beta (TGFβ) [17], platelet-derived growth factor (PDGF) [18], connective tissue growth factor (CTGF) [19], reactive oxygen species (ROS) [20], inflammation [21, 22], ECM activation (matrix metallopeptidase (MMP), tissue inhibitor of metalloprotease (TIMP)) [23], and apoptosis [24], have been implicated in atrial fibrosis and AF. Thus, the molecular determinants of atrial fibrosis have been extensively investigated, but the underlying profibrotic pathways leading to fibrosis and AF remain unclarified in patients with cardiovascular conditions.

AF usually begins in a self-terminating paroxysmal atrial fibrillation (PAF), which refers to AF that stops spontaneously within 7 days after onset, then progresses to more frequent and long-lasting episodes, and then becomes persistent AF. Atrial fibrosis has been largely regarded as a consequence of AF, but several studies have shown that increased fibrosis precedes and contributes to the development of AF in patients with sinus rhythm (SR) [25, 26]. Atrial fibrous tissue content was increased in patients with PAF [27], and LA fibrosis was described

using delayed gadolinium-enhanced cardiac magnetic resonance (DE-CMR) imaging, even in patients with PAF [28, 29]. Recent histological examinations [30] have also reported that the degree of fibrosis progressively worsened PAF to persistent AF and then permanent AF. Furthermore, Goette et al. [31] showed that PAF, as well as AF, is accompanied by elevated levels of activated extracellular signal-regulated kinase, suggesting that atrial fibrotic changes are already present in PAF. Voigt et al. [32] also reported that even during PAF, molecular pathways involved with gene transcription, such as TGFβ and PDGF, are already in play. These results suggest that atrial remodeling, including fibrosis, already exists during PAF.

LASr has been reported to be inversely correlated with the extent of atrial fibrosis and LA structural remodeling detected by DE-CMR in patients with AF [28] and atrial histological fibrosis in patients with mitral valve disease [33, 34]. Furthermore, LASr was lower in patients with PAF who subsequently developed persistent AF [35] and determined maintenance of sinus rhythm post-ablation [36]. Schaaf et al. [37] also showed that LA remodeling associated with AF is detectable by 2DSTE, even during PAF. These results suggest that LASr may be a marker of atrial fibrosis in patients with cardiovascular disease, even in patients with PAF. However, it remains unclear whether LAS is a marker of LA interstitial fibrosis and remodeling in patients with cardiovascular disease. The aim of this study was to investigate the association of LAS, atrial structural remodeling including fibrosis, and AF by analyzing gene expression in patients undergoing cardiovascular surgery.

## Methods

### Study design

Eligible patients were those undergoing planned cardiovascular surgery with cardiopulmonary bypass during thoracotomy or median sternotomy between October 13th 2015 and October 28st 2020 at Dokkyo Medical Hospital. The study protocol conformed to the Declaration of Helsinki, as approved by the institutional human research committee, and the protocol was approved by the Regional Ethics Committee of Dokkyo Medical University Hospital.

### Transthoracic echocardiography

Each patient underwent pre-operative transthoracic echocardiography. Two-dimensional (2D) images were recorded with an iE33 and EPIQ7 cardiovascular ultrasound system (PHILIPS, Amsterdam, the Netherlands) with a 1.7–3.4 MHz Doppler transducer according to the recommendations of the American Society of Echocardiography. Left atrial diameter (LAD), left ventricular end-diastolic diameter (LVDd), left ventricular end-systolic diameter (LVDs), interventricular septum thickness (IVSth), and left ventricular posterior wall thickness (PWth) were measured using a parasternal long axis view. Left ventricular end-diastolic volume (LVEDV) and end-systolic volume (LVESV) were measured from the apical view using the biplane method. Left ventricular mass (LV mass, LVM) was estimated by LVDd and wall thickness (IVSth and PWth) and then indexed to body surface area (LVMI). Left ventricular ejection fraction (LVEF) was calculated using the Simpson method.

LVEF (bp) = 100×(LVEDV-LVESV)/LVEDV

Doppler echocardiography was performed to detect E/e', the ratio of early-diastolic left ventricular inflow velocity (E) to early-diastolic mitral annular velocity (e'). LAV was measured by the biplane Simpson's method of disks. The LA was divided into a pile of disks perpendicular to the longitudinal length direction in both the 4-chamber and 2-chamber views. The radius of each disk was measured from the longitudinal axis to the LA contour in the 2 perpendicular planes. The volume of each disk was calculated automatically, and LAV was calculated by the summation of the disks' volumes as shown below:

$LAV = π/4Σ(i = 1\ to\ 20)\ ai × bi × L/N$, where ai and bi were 20 disks obtained in the 2 orthogonal incidences (the 4- and 2-chamber views). The formula for the biplane A/L method was as follows: $LAV = (8/3)\ π\ (A1 × A2)/([L1 + L2]/2)$

The LAV index (LAVI [Biplane, bp]) was obtained by dividing the LAV by the body surface area.

## 2DSTE

To obtain pre-operative LAS, 2DSTE analysis was performed using image analysis software (TOMTEC-ARENA, TomTec Imaging Systems GmbH, Munich, Germany). The analysis for LAS measurements using the intra- and inter-observer variability have been reported to show high reproducibility [38–40]. Automatic analyses (Auto Strain LA) of 4-chamber (4CH) and 2-chamber (2CH) apical view images of the LA were performed. The depth and width of the sector had been adjusted to include as little as possible outside the area of interest. Three consecutive heart cycles were recorded during a single breath hold using a frame rate of > 80 frames/second for offline analysis. The LA endocardial border was traced, and the area of interest was adjusted to include the thickness of the entire LA wall manually. Then, the software selected stable speckles within the LA wall and tracked those speckles frame-by-frame throughout the cardiac cycle. The entire LA tracking in 4CH and 2CH apical view images of the LA was divided into 6 segments by the software. If the tracking quality for each segment was not acceptable, endocardial borders were readjusted until better tracking was obtained. We could perform speckle-tracking in 118 patients (90%) participants of 131 patients. To calculate LAS, we used the upslope of the R-wave as the reference point for strain in the electrocardiogram [4]. We defined the following components of LA strain: LA reservoir strain (LASr) = peak (maximal) longitudinal LA strain; LA contractile strain (LASct) = longitudinal LA strain measured between onset of the P wave and onset of the QRS complex; and LA conduit strain (LAScd) = LASr - LASct. In patients with AF, there is no LASct owing to loss of coordinated LA contraction. A typical image of LAS obtained from 6 segments in apical 4CH view images is shown in Fig 1Aa (PAF) and 1Ab (AF). All strain values are positive. There are 2 peaks, the first corresponding to reservoir function (first peak between the R wave and the T wave), and the second corresponding to contractile function (Fig 1Aa). Furthermore, measurements of LAS were taken in the 4CH and 2CH views, and the mean value of both the 4CH and 2CH views was calculated as recommended by the European Association of Cardiovascular Imaging [41]. Two typical LA strain images from the 4CH and 2CH apical views are shown in Fig 1Ba (PAF) and 1Bb (AF). The values were averaged for all 12 segments (6 in the apical 4CH view and 6 in the apical 2CH view), and mean LASr (mLASr), LAScd (mLAScd), and LASct (mLASct) values were obtained.

## Specimen preparation

Before starting surgery, intraoperative transesophageal echocardiography was used to confirm the absence of intramural thrombi. After establishing cardiopulmonary bypass via a median sternotomy approach, the ascending aorta was cross-clamped to induce cardiac arrest. Subsequently, the heart was luxated, and the neck of the left atrial appendage was resected using a powered surgical stapler (ECHELON FLEX™ GST System, Ethicon, USA). Tissue samples were stored at -80°C and later prepared for pathological examination and molecular assessment.

## Histological staining and immunohistochemistry

Specimens were fixed with 4% paraformaldehyde phosphate buffer solution (NACALAI TES-QUE, Inc., Kyoto, Japan) and embedded in paraffin. Formalin-fixed, paraffin-embedded (FFPE) atrial tissues were sectioned at 5 μm thickness and stained with Elastica van Gieson (EVG) to visualize collagen fibers. For immunohistochemical staining, the FFPE atrial tissue

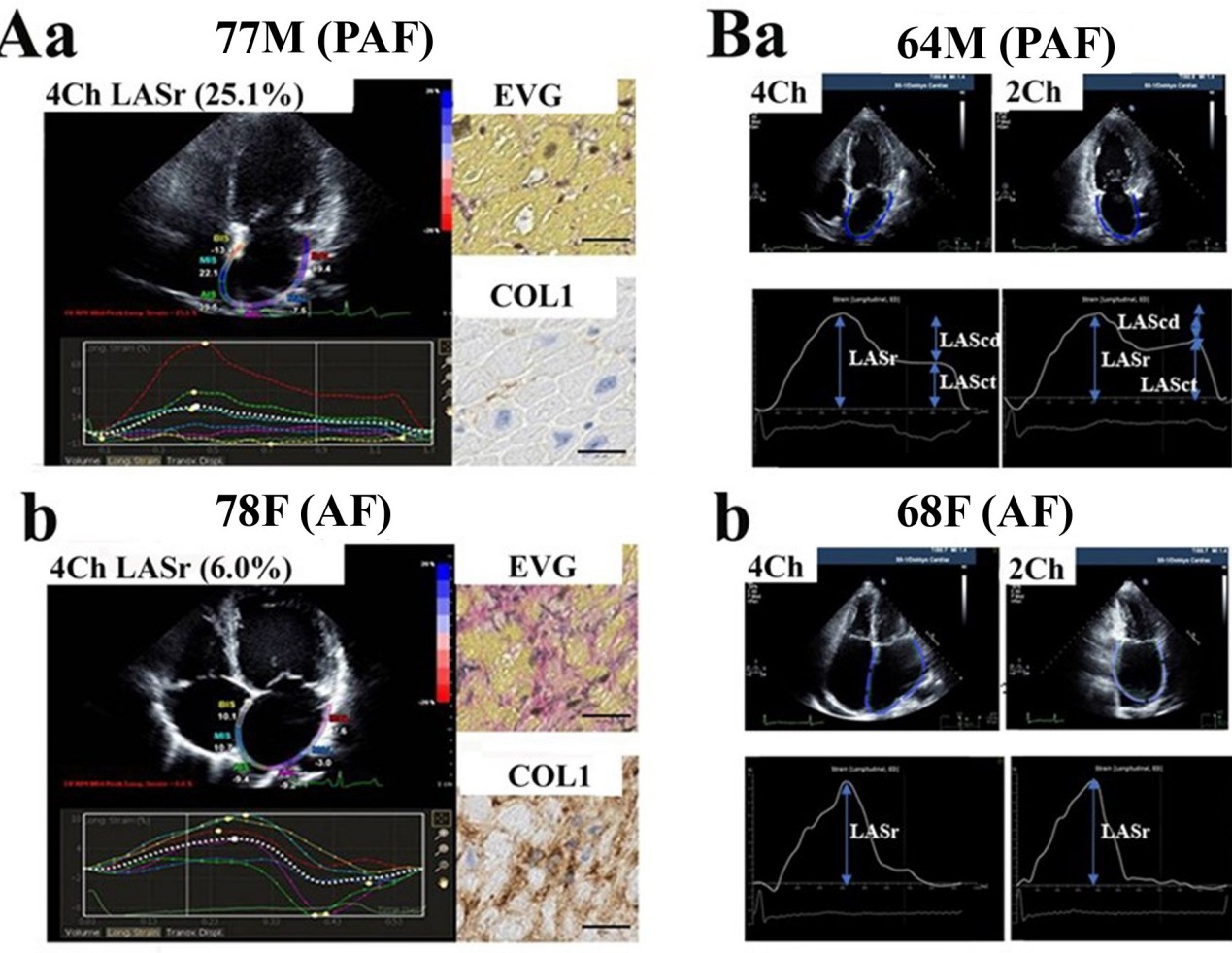

**Fig 1.** Typical left atrial strain (LAS) and histological findings in a patient with PAF (Aa, Ba) and a patient with atrial fibrillation (AF) (Ab, Bb). Aa and Ab: Quad view of longitudinal left atrial (LA) strain by 2DSTE. Elastica van Gieson stain (EVG), and immunohistochemical staining of collagen type 1a are also shown for each patient. Ba and Bb: Two typical LA strain images from 4-chamber (4CH) and 2-chamber (2CH) apical views obtained from a patient with PAF (Ba) and a patient with AF (Bb). The mean values were averaged for 6 segments in the 4CH view (left part) and 6 segments in the 2CH view (right part), as shown in the lower part. LASr, LA reservoir strain; LASct, LA contractile strain; LAScd, LA conduit strain.

sections were deparaffinized and rinsed. The sections were treated with DAKO EnVision™ FLEX Target Retrieval Solution High pH (Agilent Technologies, Inc., Santa Clara, CA, USA) at 97 ˚C for 20 min for antigen retrieval, and then with DAKO EnVision™ FLEX Peroxidase-Blocking Reagent (Agilent) for 5 min at room temperature. Then, the slides were stained using a DAKO EnVision™ FLEX System (Agilent). The first antibody reaction was performed with a 1/2000 dilution of rabbit monoclonal anti-collagen type I α1 antibody (ab138492, Abcam plc, Cambridge, UK). The specificity was controlled by adding normal rabbit IgG (I-1000, VEC-TOR LABORATORIES, Inc., Burlingame, CA, USA) as the primary antibody.

## RNA extraction and real-time quantitative reverse transcriptase/polymerase chain reaction (RT-PCR)

Cellular RNA was extracted from each LAA muscle sample using ISOGEN II (NIPPON GENE CO., LTD. Tokyo, Japan). For RT-PCR, complementary DNA (cDNA) was synthesized from

40 ng/10-μl reaction of total RNA using a ReverTra Ace qPCR RT Master Mix (TOYOBO CO., LTD. Osaka, Japan). Quantitative RT-PCR was performed using a KOD SYBR qPCR Mix (TOYOBO) and an Applied Biosystems 7300 Real-Time PCR System (Thermo Fisher Scientific, Waltham, MA, USA) as previously reported [42]. The reaction mixture was then subjected to PCR amplification with specific forward and reverse oligonucleotide primers for 40 cycles consisting of heat denaturation, annealing, and extension. PCR products were size-fractionated on 2% agarose gels and controlled under blue LED light. Primers were chosen based on the sequences of human genes, as shown in S1 Table. The RNA level was analyzed as an internal control (GAPDH) and used to normalize values for transcript quantity of mRNA. With regard to Bcl-2-associated X protein (BAX), the specific Taqman probe and primers set (Thermo Fisher Scientific) were purchased, and quantitative PCR was carried out using a THUNDERBIRD Probe qPCR Mix (TOYOBO). The analyzed genes were as follows: 1) ECM-related genes, COL1A1, encoding collagen Type I α1; COL3A1, encoding collagen Type III α1; FN1, encoding fibronectin 1; 2) ECM degradation-related genes, MMP2, encoding matrix metallopeptidase 2; TIMP1 and TIMP2, encoding tissue inhibitor of metalloprotease 1 and 2; 3) angiotensin II-related gene, ACE, encoding angiotensin I converting enzyme; 4) pro-fibrogenic cytokine-related genes, TGFB1, encoding TGFβ1; EDN1, encoding endothelin-1; PDGFD, encoding platelet derived growth factor D; CTGF, encoding connective tissue growth factor; 5) oxidant stress-related genes, NOX2 and NOX4 encoding NADPH oxidase 2 and 4; 6) inflammation-related genes, IL1B, encoding interleukin 1β; TNF, encoding TNFα; NLRP3, encoding NLR family pyrin domain containing 3; MCP-1, encoding monocyte chemoattractant protein 1; 7) apoptosis-related gene, BAX, encoding BCL2-associated X protein; 8) Natriuretic Peptide A-related gene, NPPA, encoding Atrial Natriuretic Peptide (ANP); and 9) GAPDH gene, encoding glyceraldehyde-3-phosphate dehydrogenase as an internal control, as shown in S1 Table.

## Statistical analysis

All data are presented as the mean ± standard deviation, median and interquartile range, or proportion, depending on their distributions. After testing for normality examination (Kolmogorov–Smirnov test or Shapiro–Wilk test), the comparisons of means between groups were analyzed with a two-sided, unpaired Student's t-test in the case of normally distributed parameters or with the Mann–Whitney-U-test in the case of non-normally distributed parameters. Associations among parameters were evaluated with Pearson or Spearman correlation coefficients. The strength of correlation was set as follows: strong ($R \geqq 0.75$), moderate ($0.5 \leqq R < 0.75$) and weak ($0.5 < R$). Receiver operating characteristic curve (ROC) analysis was performed to identify optimal cut-off levels of echocardiographic parameters for detecting the presence or absence of AF. Univariate regression analysis with COL1A1 expression level as the dependent variable and fibrosis-related gene expression as the independent variable was performed in the total population of patients, and the adjusted R-squared value ($R^2$) and $P$-value were obtained. Furthermore, multiple linear regression analyses with COL1A1 expression level as the dependent variable were performed to identify factors independently associated with echocardiographic parameters (eg, mLASr) or mRNA expression level (ie, TGFB1, NOX4). Age, sex, and body mass index (BMI) were employed as covariates in the total population of patients. When the residuals between the predicted values calculated from the prediction equation and the actual measured values for both the dependent and independent variables did not follow a normal distribution, the data were transformed logarithmically to achieve a normal distribution. Analyses were performed with SPSS version 26 (IBM Corp., New York, USA) for Windows. $P < 0.05$ was regarded as significant.

## Results

### Patients

One hundred thirty-one patients (78 males [60%]; mean age, 72.0 [64.0–79.0] years) were enrolled in the study. Twenty-two underwent CABG. Thirty-two underwent aortic valve replacement (AVR) for aortic stenosis (AS, n = 28) or aortic regurgitation (AR, n = 4). Thirty-one had mitral valve replacement (MVR) or plasty (MVP) for mitral regurgitation (MR, n = 29), and mitral stenosis (n = 2). Thirty-nine of the 131 enrolled patients (30%) had AF (Table 1). All patients had undergone medical treatments, as shown in Table 1. Fifty-nine patients received LA resection are also shown in Table 1. Twenty-four patients had MR and underwent MVR or MVP. Ten patients had AS and underwent AVR. Three patients underwent coronary artery bypass grafting (CABG), and 7 received CABG combined with valve replacement or plasty. Four patients had AVR combined with another valve replacement or plasty. Among the 59 patients who underwent LA resection, 24 had experienced PAF that stopped spontaneously within 7 days after onset and had SR on admission and during cardiac surgery. Thirty-five patients had AF, as shown in Table 1. Thirty-five patients were males. The mean (range) age was 72.0 (63.0–78.0) years. The baseline characteristics of the patients, including their risk factors, medical treatments, and other operative procedures, are shown in Table 1. The etiology of MR (rheumatic, degenerative, prolapse, and secondary) and the ratio of etiology of MR are also described in Table 1.

### Correlations between LAS and echocardiographic parameters

The LAVI (bp) was $50.9 \pm 25.5$ ml/m$^2$ in the total population of patients and $62.4 \pm 23.7$ ml/m$^2$ in patients who underwent LA resection. Detailed LAS data with averaged values for all 6 segments in the apical 4CH view and in the apical 2CH view are also presented in Table 1. mLASr and mLAScd were 16.0% (11.8%–25.8%) and $11.6\% \pm 5.6\%$, respectively, in the total population of patients. mLASr and mLAScd were 13.9% (10.2%–18.7%) and 11.7% (9.2%–14.2%), respectively in patients who underwent LA resection. The mLASct value was $10.4\% \pm 7.0\%$ and $6.9\% \pm 5.5\%$ in SR patients including PAF and in PAF patients who underwent LA resection, respectively. Fig 2 and S2 Table show the relationships between LAS and echocardiographic parameters. Neither mLASr nor mLASct correlated significantly with age or sex (S2 Table). mLASr weekly positively correlated with BMI, and mLAScd weekly correlated with age and sex in the total population of patients. mLASr correlated with mLAScd ($R = 0.472$, $P<0.001$, Fig 2A) in the total population of patients and mLAScd ($R = 0.543$, $P<0.001$) and mLASct ($R = 0.609$, $P<0.001$, Fig 2B) in SR patients including PAF. mLASr strongly negatively correlated well with LAVI (bp) ($R = -0.736$, $P<0.001$, Fig 2C) in the total population of patients and moderately correlated with LAVI (bp) in SR patients including PAF ($R = -0.589$, $P<0.001$). mLASr weakly negatively correlated with E/e' ($R = -0.216$, $P = 0.041$, Fig 2D) and positively LVEF (bp) ($R = 0.312$, $P = 0.004$, Fig 2E) in the total population of patients. mLAScd negatively weekly correlated with LAVI (bp) ($R = -0.236$, $P = 0.036$, Fig 2F). No correlation was observed between each individual LAS parameter and cardiovascular risk factors or drug use (S3 Table).

### Echocardiographic differences between patients with AF and SR including PAF

Fig 3A shows the echocardiographic differences between patients with AF and SR patients including PAF. Patients with AF had higher LAD (52.7 [43.5–59.9] vs. 41.0 [36.3–45.8] mm, respectively; $P<0.001$) and LAVI (bp) (70.7 [62.8–79.4] vs. 37.4 [24.6–50.5] ml/m$^2$,

**Table 1. Characteristics of the patients.**

| | All patients | Patients who underwent LA resection |
|---|---|---|
| Number | 131 | 59 |
| Male / Female, n (%) | 78 (60) / 53 (40) | 35 (59) / 24 (41) |
| Age, y, median (range) | 72.0 (64.0–79.0) | 72.0 (63.0–78.0) |
| BMI, kg/m$^2$, median (IQR) or median (range) | 22.9 (20.8–25.9) | 22.6 ± 3.9 |
| Atrial fibrillation, n (%) | 39 (30) | 35 (59) |
| NYHA class, median (IQR) | 2.0 (1.0–3.0) | 2.0 (2.0–3.0) |
| Risk factors, n (%) | | |
| Hypertension | 90 (69) | 41 (69) |
| Diabetes | 36 (27) | 10 (17) |
| Dyslipidemia | 58 (44) | 19 (32) |
| Chronic kidney disease | 68 (52) | 34 (58) |
| Cardiovascular surgery, n (%) | | |
| CABG | 22 (17) | 3 (5) |
| AS (AVR) | 28 (21) | 10 (17) |
| AR (AVR) | 4 (3) | 3 (5) |
| MR (MVR, MVP) | 29 (22) | 24 (41) |
| MS (MVR) | 2 (2) | 2 (3) |
| CABG combined with valve procedure (AVR, MVP, MVR) | 13 (10) | 7 (12) |
| AVR(AS, AR) with MVR or MVP(MR) | 7 (5) | 4 (7) |
| Aortic disease (AAR, TAR, HAR, et al) | 9 (7) | 0 (0) |
| Other | 17 (13) | 6 (10) |
| MR etiology, n (%) | | |
| Rheumatic / degenerative / prolapse / secondary (ischemic) / others, n (%) | 5 (10) / 8 (16) / 24 (49) / 9 (18) / 3 (7) | 5 (14) / 7 (20) /16 (46) / 6 (17) /1(3) |
| Drug use, n (%) | | |
| β-blockers | 72 (55) | 34 (58) |
| Ca$^{2+}$-blockers | 48 (37) | 23 (39) |
| vACE-I/ARB | 78 (60) | 38 (64) |
| Statins | 58 (44) | 19 (32) |
| Anti-diabetic drugs | 29 (22) | 10 (17) |
| eGFR, ml/min/1.73 m$^2$, median (IQR) | 59.6 (44.8–76.0) | 58.9 (44.3–70.8) |
| BNP, pg/ml, median (IQR) | 217.0 (74.6–491.6) | 300.6 (152.1–526.7) |
| HbA1c, %, median (IQR) | 5.9 (5.5–6.6) | 5.8 (5.6–6.4) |
| hsCRP, mg/L, median (IQR) | 0.13 (0.04–0.50) | 0.12 (0.06–0.56) |
| Echocardiographic data, median (IQR) or median (range) | | |
| LAD, mm | 42.4 (37.8–49.0) | 48.4 ± 8.6 |
| LVDd, mm | 50.0 (44.7–59.8) | 54.5 ± 10.8 |
| LVDs, mm | 34.4 (28.8–41.5) | 37.9 ± 10.3 |
| LVEF (bp), % | 61.0 (52.0–66.0) | 62.0 (52.0–65.0) |
| LVMI, g/m$^2$ | 112.9 ± 38.9 | 114.0 (98.0–141.0) |
| E/e' | 18.1 (13.1–24.5) | 19.9 (15.8–25.9) |
| LAVI (bp), ml/m$^2$ | 50.9 ± 25.5 | 62.4 ± 23.7 |
| LASr 4CH, % | 16.3 (12.0–25.3) | 13.8 (10.2–20.9) |
| LASr 2CH, % | 16.9 (10.8–26.4) | 12.9 (8.5–18.5) |
| Mean LASr (mLASr), % | 16.0 (11.8–25.8) | 13.9 (10.2–18.7) |
| LAScd 4CH, % | 11.7 ± 5.9 | 12.0 (8.8–14.8) |
| LAScd 2CH, % | 10.4 (7.2–14.5) | 11.4 ± 5.6 |

*(Continued)*

**Table 1.** (Continued)

| | All patients | Patients who underwent LA resection |
|---|---|---|
| Mean LAScd (mLAScd), % | 11.6 ± 5.6 | 11.7 (9.2–14.2) |
| LASct 4CH, % | 10.3 ± 7.1* | 7.2 ± 5.6* |
| LASct 2CH, % | 10.9 ± 8.1* | 6.6 ± 6.5* |
| Mean LASct (mLASct), % | 10.4 ± 7.0* | 6.9 ± 5.5* |

Data are shown as mean ± standard deviation, or, where indicated, median and interquartile range. Categorical variables are expressed as number and percentage. LA, left atrium; BMI, body mass index; NYHA, New York Heart Association; CABG, coronary artery bypass grafting; AS, aortic stenosis; AVR, aortic valve replacement; AR, aortic regurgitation; MR, mitral regurgitation; MVR, mitral valve replacement; MVP, mitral valve plasty; AAR, ascending aorta replacement; TAR, total arch replacement; HAR, hemiarch replacement; ACE-I, angiotensin converting enzyme inhibitor; ARB, angiotensin II receptor blocker; Anti-diabetic drugs (i.e. α-glucosidase inhibitor, sulfonylurea, biguanide, dipeptidyl peptidase-4 inhibitor, sodium glucose cotransporter 2 inhibitor); eGFR, estimated glomerular filtration rate; BNP, brain natriuretic peptide; hsCRP, high-sensitivity C-reactive protein; HbA1c, hemoglobin A1c; LAD, left atrial diameter; LVDd, left ventricular end-diastolic diameter; LVDs, left ventricular end-systolic diameter; LVEF (bp), LV ejection fraction measured by the biplane Simpson method; LVMI, LV mass index; E/e', the ratio of early-diastolic left ventricular inflow velocity (E) to early-diastolic mitral annular velocity (e'); LAVI (bp), left atrial volume index estimated by the biplane area-length method; LAS, left atrial longitudinal strain (r, reservoir; cd, conduit; ct, contraction); 4CH, 4 chamber; 2CH, 2 chamber; Mean value = (4CH value + 2CH value) / 2
*exclusion of patients with atrial fibrillation.

respectively; $P<0.001$), compared with SR patients including PAF. mLASr was significantly lower in patients with AF than in SR patients with PAF (10.4 [8.3–13.4] % vs., 22.5 [15.2–29.0] %, respectively; $P<0.001$). No significant differences were observed between patients with AF and SR including PAF in LVEF, E/e' and mLAScd. Next, we constructed an ROC plot to identify the optimal cut-off levels of echocardiographic parameters to detect the presence of AF

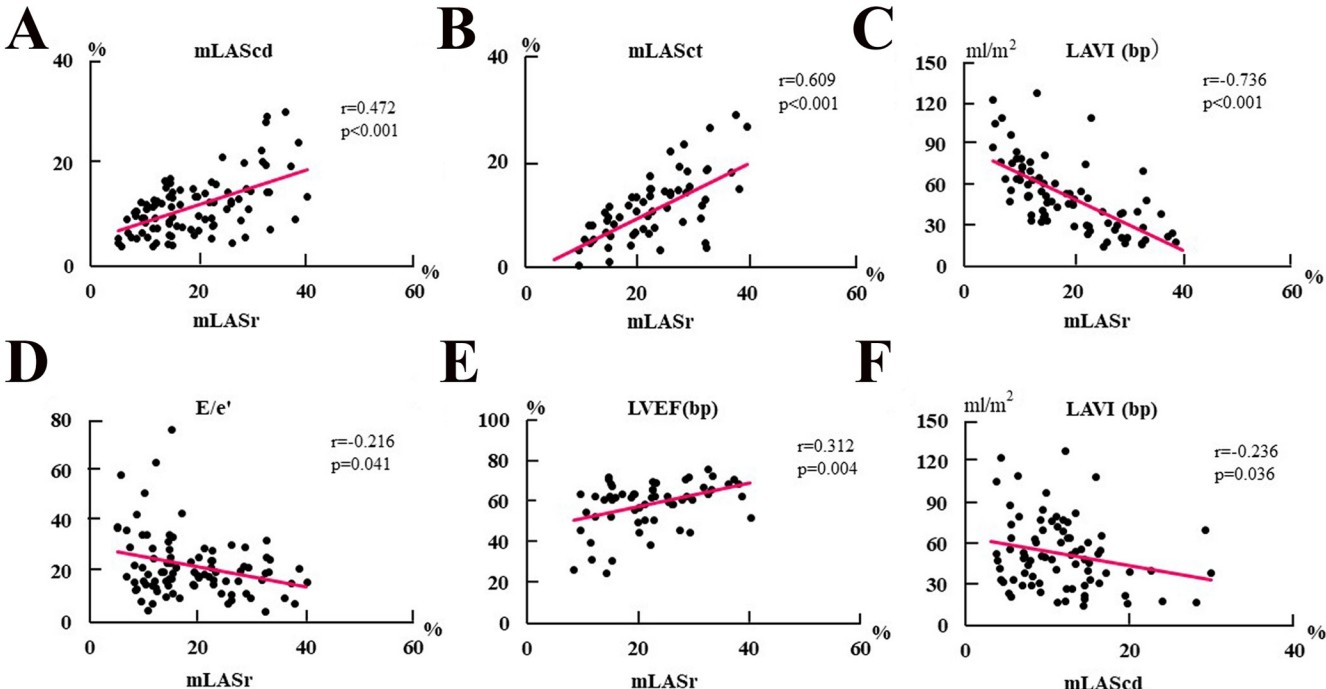

**Fig 2. Relationships between left atrial strain (LAS) and echocardiographic findings (LAVI (bp), E/e', LVEF (bp)).** The data of mLASr and mLAScd were obtained from 118 patients including PAF and AF. The data of mLASct were obtained from 84 patients with SR patients including PAF. mean LASr value (mLASr), LAScd (mLAScd), LASct (mLASct)). *R*- and *P*-value are shown. *$P<0.05$, **$P<0.01$, ***$P<0.001$.

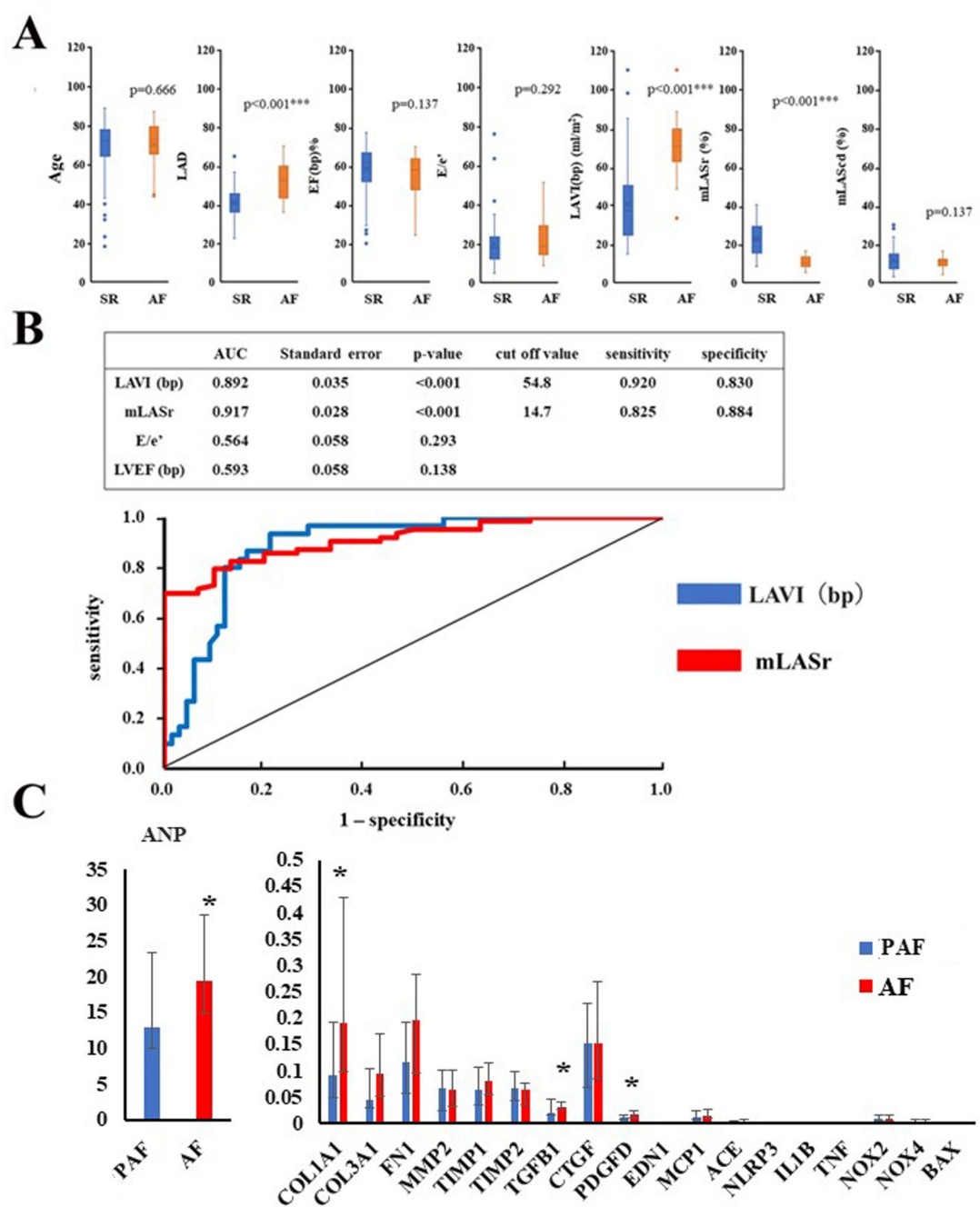

**Fig 3. Comparison of echocardiographic parameters and gene expression level between patients with SR including PAF and those with AF.** A: Comparison of echocardiographic parameters. SR including PAF (blue, n = 84), AF (red, n = 34). B: ROC to identify the optimal cut-off level of echocardiographic findings to detect AF (n = 118). C: Comparison of ANP, and fibrosis-related gene expression level in between patients with PAF (n = 23–24) and those with AF (n = 33–35). PAF (blue), AF (red). *P<0.05, ***P<0.001.

(Fig 3B). To generate the ROC plot, different echocardiographic parameter cut-off values were used with true-positives on the vertical axis (sensitivity) and false-positives (1-specificity) on the horizontal axis. The area under the curve (AUC) of mLASr was 91.7%. Sensitivity and specificity were 82.5% and 88.4%, respectively. The optimal cut-off value was 14.7%. The AUC

of the LAVI (bp) was 89.2%. Sensitivity and specificity were 92.0% and 83.0%, respectively. The optimal cut-off value was 54.8 ml/m². Neither E/e' nor LVEF (bp) predicted AF presence.

## Correlations with atrial gene expression level

We investigated relationships between the ANP mRNA expression level and atrial fibrosis-related gene expression. Fig 4A shows the correlations between ANP and various mRNA expression levels in the total population of patients. Significant correlations were observed between ANP mRNA and ECM-related genes (COL1A1, COL3A1, FN1), NOX4, TIPM1, pro-fibrogenic cytokine genes (TGFB1, EDN1, CTGF, PDGFD), NLRP3, and BAX mRNA. The relationship between ANP and COL1A1 mRNA expression is shown in Fig 4B. A significant correlation was positively observed ($R = 0.365$, $P = 0.005$). Fig 4A also shows correlations among various mRNA expression levels (heat map) in all patients. Interstitial fibrosis (primarily composed of collagen) can be a critical element of atrial fibrosis. Therefore, we assessed the relationships between COL1A1 mRNA expression and atrial fibrosis-related gene expression. The COL1A1 expression level strongly positively correlated with other ECM-related genes, such as COL3A1 (Fig 4B, $R = 0.939$, $P<0.001$) and FN1 (Fig 4B, $R = 0.926$, $P<0.001$). It also positively correlated with ECM degradation-related genes (MMP2 ($R = 0.716$, $P<0.001$), TIMP1 ($R = 0.795$, $P<0.001$), TIMP2 ($R = 0.725$, $P<0.001$), pro-fibrogenic cytokines (TGFB1 [Fig 4B, $R = 0.853$, $P<0.001$], EDN1 [$R = 0.704$, $P<0.001$], PDGFD [$R = 0.688$, $P<0.001$], CTGF [$R = 0.646$, $P<0.001$]), oxidant stress-related genes (NOX2 [$R = 0.767$, $P<0.001$], NOX4 [$R = 0.812$, $P<0.001$]), ACE ($R = 0.685$, $P<0.001$), inflammation-related genes (NLRP3 [$R = 0.664$, $P<0.001$], IL1B [$R = 0.611$, $P<0.001$], MCP-1 [$R = 0.562$, $P<0.001$], apoptosis-related gene (BAX [$R = 0.555$, $P<0.001$]), and weekly TNF [$R = 0.460$, $P<0.01$]) as shown in Fig 4. Similar relationships between COL1A1 expression levels and ANP mRNA and fibrosis-related gene expression (COL3A1, FN1, TGFB1) were observed in PAF patients, as shown in Fig 4C. The expression level of COL1A1 mRNA weekly positively correlated with that of ANP (Fig 4C, $R = 0.483$, $P<0.001$), and strongly positively correlated COL3A1 (Fig 4C, $R = 0.867$, $P<0.001$), FN1 (Fig 4C, $R = 0.859$, $P<0.001$), and TGFB1 mRNA (Fig 4C, $R = 0.839$, $P<0.001$).

Furthermore, significant correlations were observed among ECM-related genes (COL1A1, COL3A1, FN1), ECM degradation-related genes (MMP2, TIMP1, TIMP2), oxidant stress-related genes (NOX2, NOX4), pro-fibrogenic cytokines (TGFB1, EDN1, PDGFD, CTGF), ACE, and inflammation-related genes (IL1B, TNF, NLRP3, MCP-1), and apoptosis (BAX), as shown in Fig 4A. The mRNA levels of ECM-related genes (COL3A1 and FN1) as well as COL1A1 each positively correlated with atrial fibrosis-related gene expression. These results suggest that multiple signaling mechanisms are involved and linked in atrial fibrosis, reflected as an increase in ECM gene expression (COL1A1, COL3A1, FN1).

## Relationships between the expression level of COL1A1 and other fibrosis-related genes

Univariate regression analysis with log (COL1A1) expression level as the dependent variable and fibrosis-related gene expression as the independent variable was performed in all patients, and the adjusted R-squared value ($R^2$) and $P$ value were obtained. The adjusted $R^2$ value for each gene is shown in Fig 5. The genes identified with an adjusted $R^2$ above 0.5 were those encoding a pro-fibrogenic cytokine (TGFB [adjusted $R^2 = 0.622$, $P<0.001$]), oxidant stress (NOX4 [adjusted $R^2 = 0.622$, $P<0.001$]), NOX2 (adjusted $R^2 = 0.611$, $P<0.001$), ECM degradation-related genes (TIMP1 [adjusted $R^2 = 0.597$, $P<0.001$], MMP2 [adjusted $R^2 = 0.564$, $P<0.001$], TIMP2 [adjusted $R^2 = 0.518$, $P<0.001$]), an inflammation-related gene (NLRP3 [adjusted $R^2 = 0.517$, $P<0.001$]) and EDN1 (adjusted $R^2 = 0.506$, $P<0.001$). Thus, TGFB1 was

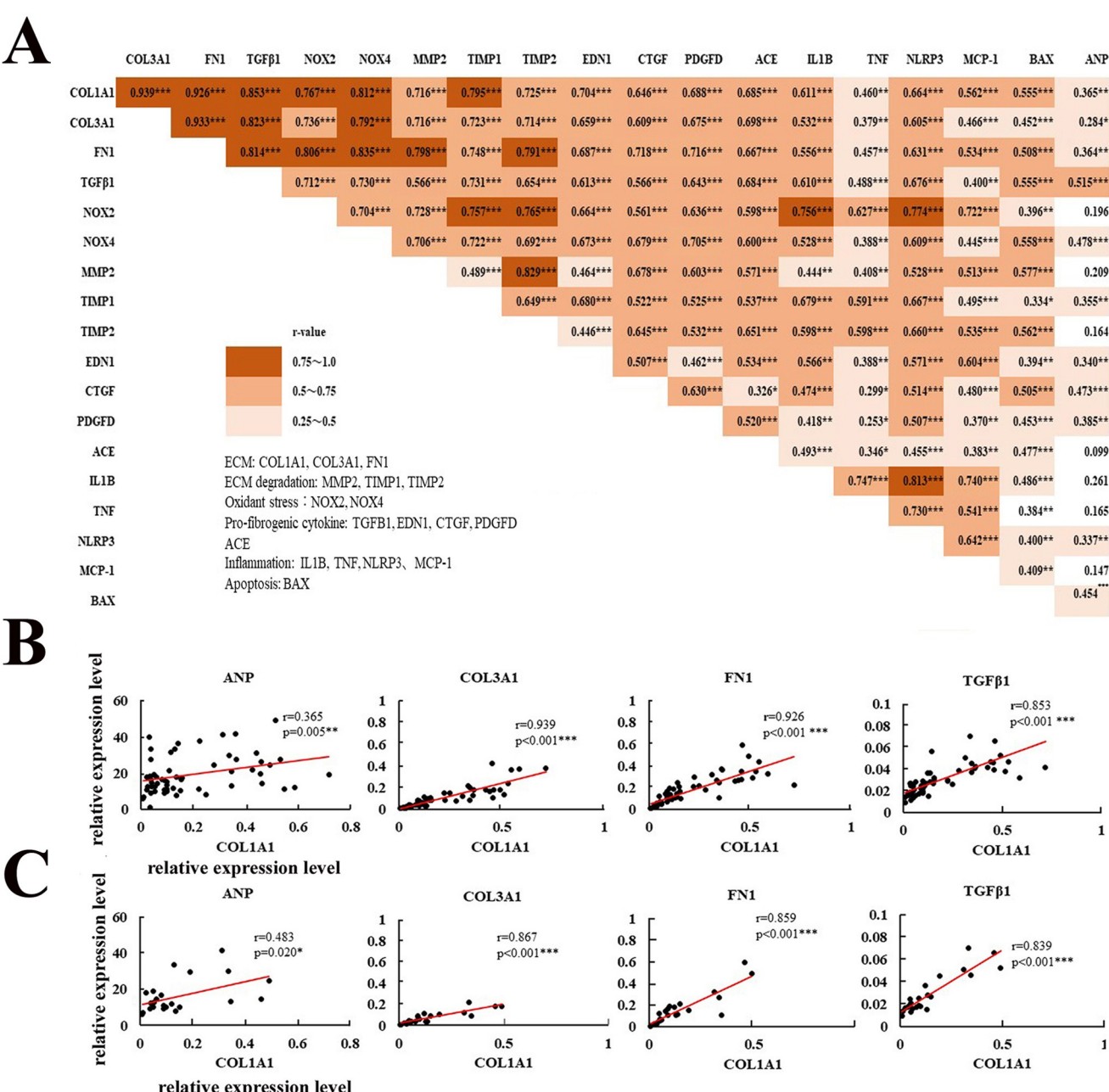

**Fig 4. Correlations among atrial gene expression level including ANP, and fibrosis-related gene expression in total patients.** A: Correlations among atrial gene expression level including ANP, and fibrosis-related gene expression in total patients receiving LA resection (n = 57–59). The *R*-value and *P*-value among genes are indicated. *$P < 0.05$, **$P < 0.01$, ***$P < 0.001$. The orange area shows statistical significance. B: Correlations between COL1A1 and fibrosis-related gene expression level (ANP, COL3A1, FN1, TGFB1) in total patients receiving LA resection (n = 57–59). C: Correlations between COL1A1 and fibrosis-related gene expression level (ANP, COL3A1, FN1, TGFB1) in PAF patients (n = 23–24). *$P < 0.05$, **$P < 0.01$, ***$P < 0.001$.

the strongest independent variable for predicting the log (COL1A1) expression level. The oxidant stress (NOX4/NOX2), and ECM degradation-related genes (TIMP1, TIMP2/MMP2) were the next strongest predictors of log (COL1A1) expression level among the genes examined in this study. Furthermore, multiple linear regression analysis with log (COL1A1) expression level as the dependent variable and fibrosis-related gene expression (TGFB1, NOX4, and

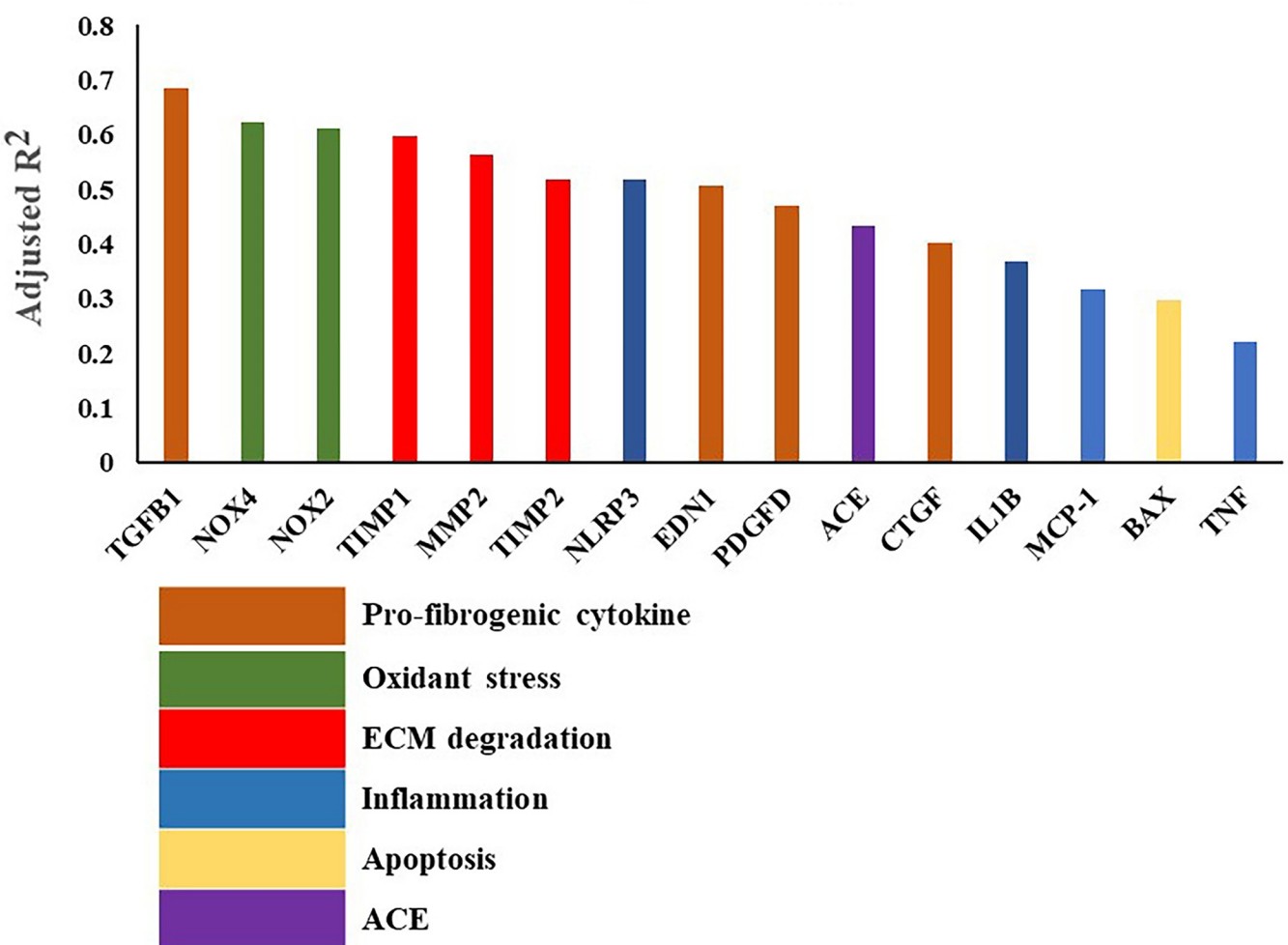

**Fig 5. The univariate regression analysis with log (COL1A1) expression level as the dependent variable and fibrosis-related gene expression as the independent variable.** The adjusted $R$-squared value ($R^2$) is shown. The data were obtained in 57–59 patients receiving LA resection. Pro-fibrogenic cytokine genes (TGFB1, EDN1, PDGFD, CTGF), oxidant stress-related genes (NOX2, NOX4), ECM degradation-related genes (MMP2, TIMP1, TIMP2), inflammation-related genes (NLRP3, IL1B, MCP-1, TNF), angiotensin-related gene (ACE), apoptosis-related gene (BAX). All data are P<0.001.

TIMP1) selected in Fig 5 as the independent variable was performed in the total population of patients. Findings showed that log (TGFB1) (β = 0.415, P<0.001), log (NOX4) (β = 0.346, P<0.001) and log (TIMP1) (β = 0.231, P = 0.022) were independent variables to predict the log (COL1A1) expression level. The analysis also showed that log (TGFβ1) (β = 0.463, P<0.001) and log (NOX4) (β = 0.349, P<0.001) were independent determinants of COL1A1 expression levels, even after adjusting for age, sex, and BMI. To predict the log (COL1A1) expression level, TGFB1 and NOX4 expression levels were used and calculated as follows: log (COL1A1) = 1.280 × log (TGFB1) + 0.629 × log (NOX4) + 6.979 (n = 58, $R^2$ = 0.794, P<0.001).

## Relationships between echocardiographic parameters and fibrosis-related gene expression levels

We investigated relationships between echocardiographic parameters and COL1A1 mRNA expression. As shown in Fig 6, the COL1A1 expression level correlated with sex in the total population of patients. A week but significant negative correlation was observed between

**Fig 6. Relationships between COL1A1 mRNA expression level and echocardiographic parameters.** A: Relationships between COL1A1 mRNA expression and clinical data (age, sex, BMI) and echocardiographic parameters in PAF patients (n = 20) and total patients (n = 52) receiving LA resection. *R*- and *P*–values are shown. *P<0.05 B and C: Relationships between COL1A1 mRNA expression level and LAVI (bp) or mLASr in total patients (B, n = 52) and PAF patients (C, n = 20) receiving LA resection. *P<0.05.

mLASr and the COL1A1 expression level (*R* = –0.309, *P* = 0.039, Fig 6B). Neither mLAScd nor mLASct correlated significantly with COL1A1. Neither LAD, LAVI (bp) (*R* = 0.217, *P* = 0.138, Fig 6B) nor E/e' correlated with COL1A1 gene expression in the total population of patients. Similarly, in PAF patients, moderate negative correlations between mLASr and the COL1A1

gene expression level ($R$ = −0.539, $P$ = 0.017, Fig 6C) were observed. Neither LAD, LAVI (bp) ($R$ = 0.352, $P$ = 0.128, Fig 6C) nor E/e' correlated with COL1A1 gene expression. Multiple regression analysis with log (COL1A1) mRNA expression level as the dependent variable and echocardiographic findings (log (mLASr)) as the independent variable was performed in PAF patients. Results showed that mLASr was an independent variable for predicting the COL1A1 mRNA expression level (β = −0.593, $P$ = 0.029 adjusted for age, β = −0.547, $P$ = 0.025 adjusted for sex). Similarly, in the total population of patients, multiple regression analysis showed that log (mLASr) was an independent predictor of log (COL1A1) expression (β = −0.355, $P$ = 0.030), even after adjusting for age, sex, and BMI. These results suggest that mLASr is a predictor of the atrial COL1A1 expression level, even in PAF patients.

## Comparison of gene expression levels between PAF and AF patients

Fig 3C shows comparisons of gene expression levels between PAF and AF patients. The expression of ANP ($P$ = 0.016), COL1A1 ($P$ = 0.031), TGFB1 ($P$ = 0.030) and PDGFD ($P$ = 0.027) mRNA significantly increased in patients with AF compared with patients with SR. This is consistent with EVG staining to visualize collagen fibers, and immunohistochemical staining of collagen type I α1 (Fig 1), as marked staining of collagen with EVG staining and collagen type I α1 with immunohistochemical staining was observed in patients with AF compared with PAF patients. However, the expression levels of other genes, including COL3A1, FN1, and NOX2, were not statistically different between PAF and AF patients. Therefore, we compared the gene expression level between PAF with low mLASr and high mLASr, as shown in Table 2. Patients with PAF were divided into 2 groups based on the median value of mLASr. PAF patients with low mLASr had low mLAScd, compared with PAF patients with high mLASr. However, both groups had similar LAD, LAVI (dp), mLASct, and E/e'. Patients with low mLASr showed higher expression of ANP, ECM-related genes (COL1A1, COL3A1, FN1), ECM degradation-related genes (TIMP1, TIMP2), pro-fibrogenic cytokines (TGFB1, PDGFD, EDN1), NLRP3, oxidant stress-related genes (NOX2, NOX4), and BAX mRNA, compared with those with high mLASr, as shown in Table 2.

Furthermore, we investigated the correlation matrix between echocardiographic findings and atrial gene expression levels in PAF patients (A) and AF patients (B), as shown in Fig 7. In PAF patients, mLASr negatively moderately correlated with extracellular matrix (COL1A1 ($R$ = −0.539, $P$<0.05), COL3A1 ($R$ = −0.642, $P$<0.01), and FN1 ($R$ = −0.584, $P$<0.01) mRNA expression level. mLASct also negatively correlated with COL3A1 ($R$ = −0.542, $P$<0.05), and FN1 ($R$ = −0.488, $P$<0.05), but not with COL1A1. mLAScd did not significantly correlate with ECM genes. In addition, mLASr was negatively correlated with TGFB1, NOX2, NOX4, TIMP1, TIMP2, EDN1, and NLRP3 in PAF patients. On the other hand, in AF patients, neither mLASr nor mLAScd was significantly correlated with the expression of ECM-related genes (COL1A1, COL3A1, FN1), and other fibrosis-related genes. These results suggest that mLASr correlates well ECM-related gene (COL1A1, COL3A1, FN1) expression levels and fibrosis-related gene expression levels in PAF patients but not in patients with AF.

## Discussion

LA remodeling is due to altered atrial structure following pressure and volume overload, leading to AF [3]. A long duration of AF increases LA size and volume, and the LAVI is used as a predictor of AF and congestive heart failure [4, 43]. A LAVI > 40 ml/m$^2$ is a predictor of AF and mortality or need for mitral surgery in patients with organic MR [44]. We showed that among patients undergoing cardiovascular surgery, the LAD and LAVI were higher in the AF group than in the SR group. The AUC and optimal cut-off value of LAVI (bp) to predict AF

**Table 2. Comparison of gene expression levels and echocardiographic parameters between PAF patients with low mLASr and high LASr.**

| | High mLASr | Low mLASr | P value |
|---|---|---|---|
| Age, years | 62.6 ± 14.8 | 73.5 ± 6.0 | 0.052 |
| Sex, M/F | 7 / 3 | 5 / 5 | 0.374 |
| BMI, kg/m$^2$ | 23.1 ± 5.5 | 21.9 ± 3.8 | 0.581 |
| mLASr | 27.7 (23.0–32.6) | 15.0 (11.9–16.3) | **<0.001**\*\*\* |
| mLAScd | 16.0 (12.1–24.2) | 9.5 (6.2–13.7) | **0.006**\*\* |
| mLASct | 9.4 (5.5–13.8) | 5.2 (2.6–6.8) | 0.074 |
| LAD | 44.0 (37.8–51.3) | 47.4 (42.0–49.0) | 0.651 |
| LAVI (bp) | 40.4 (24.0–73.2) | 52.3 (41.1–72.1) | 0.588 |
| LVEF (bp) | 65.0 (49.3–69.5) | 61.0 (51.0–64.0) | 0.416 |
| E/e' | 19.0 (15.0–22.5) | 21.9 (19.3–34.2) | 0.137 |
| COL1A1 | 0.067 (0.032–0.140) | 0.312 (0.107–0.404) | **0.015**\* |
| COL3A1 | 0.037 (0.015–0.082) | 0.116 (0.062–0.182) | **0.008**\*\* |
| FN1 | 0.089 (0.025–0.148) | 0.185 (0.109–0.408) | **0.035**\* |
| MMP2 | 0.046 (0.009–0.077) | 0.075 (0.054–0.112) | 0.095 |
| TIMP1 | 0.036 (0.022–0.065) | 0.106 (0.065–0.259) | **0.008**\*\* |
| TIMP2 | 0.062 (0.028–0.071) | 0.099 (0.077–0.107)] | **0.016**\* |
| TGFB1 | 0.018 [0.014–0.027] | 0.046 [0.021–0.059] | **0.016**\* |
| CTGF | 0.095 [0.060–0.213] | 0.215 [0.143–0.338] | 0.070 |
| PDGFD | 0.011 (0.005–0.013) | 0.015 (0.011–0.022) | **0.041**\* |
| END1 | 0.0009 (0.0005–0.0012) | 0.0013 (0.0008–0.0026) | **0.044**\* |
| MCP-1 | 0.015 (0.006–0.024) | 0.017 (0.011–0.028) | 0.458 |
| ACE | 0.003 (0.002–0.006) | 0.006 (0.003–0.011) | 0.151 |
| NLRP3 | 0.0009 (0.0003–0.0015) | 0.0022 (0.0018–0.0033) | **0.014**\* |
| IL1B | 0.0005 (0.0002–0.0008) | 0.0009 (0.0006–0.0012) | 0.087 |
| TNF | 0.0003 (0.0002–0.0007) | 0.0007 (0.0003–0.0012) | 0.208 |
| NOX2 | 0.006 (0.001–0.010) | 0.017 (0.008–0.024) | **0.024**\* |
| NOX4 | 0.003 (0.002–0.004) | 0.008 (0.006–0.013) | **0.008**\*\* |
| BAX | 0.0002 (0.0001–0.0002) | 0.0003 (0.0002–0.0005) | **0.047**\* |
| ANP | 10.1 (7.5–13.7) | 21.4 (12.9–33.2) | **0.008**\*\* |

Data are shown as mean ± standard deviation, or, where indicated, median and interquartile range.

\*$P<0.05$

\*\*$P<0.01$

\*\*\*$P<0.001$.

were 88.2% and 54.8 ml/m$^2$, respectively. Alternatively, mLASr was significantly lower in the AF group, and the AUC and the optimal cut-off value of LASr to predict AF were 88.3% and 15.55%, respectively. The value of The AUC was quite near that of the LAVI (bp). LVEF and E/e' failed to predict AF. In our control data using elderly subjects without any organic heart diseases including valvular diseases (109 subjects, mean age 72 years, 57 males, unpublished data), mLASr, mLAScd, LASct and LAVI (bp) were 36.0% (33.6–39.3), 17.3% (15.0–21.6), 18.1% (15.2–21.0), and 25.6 ± 6.6, respectively. Thus, the patients studied in the present study had much lower LASr, LAScd, and LASct, and high LAVI (bp), compared with the control subjects. In healthy individuals, LASr and LAScd have been reported to decrease with age [45, 46], which was confirmed in the control subjects (data not shown). However, in the present study of patients undergoing cardiovascular surgery, mLAScd value decreased with age, but

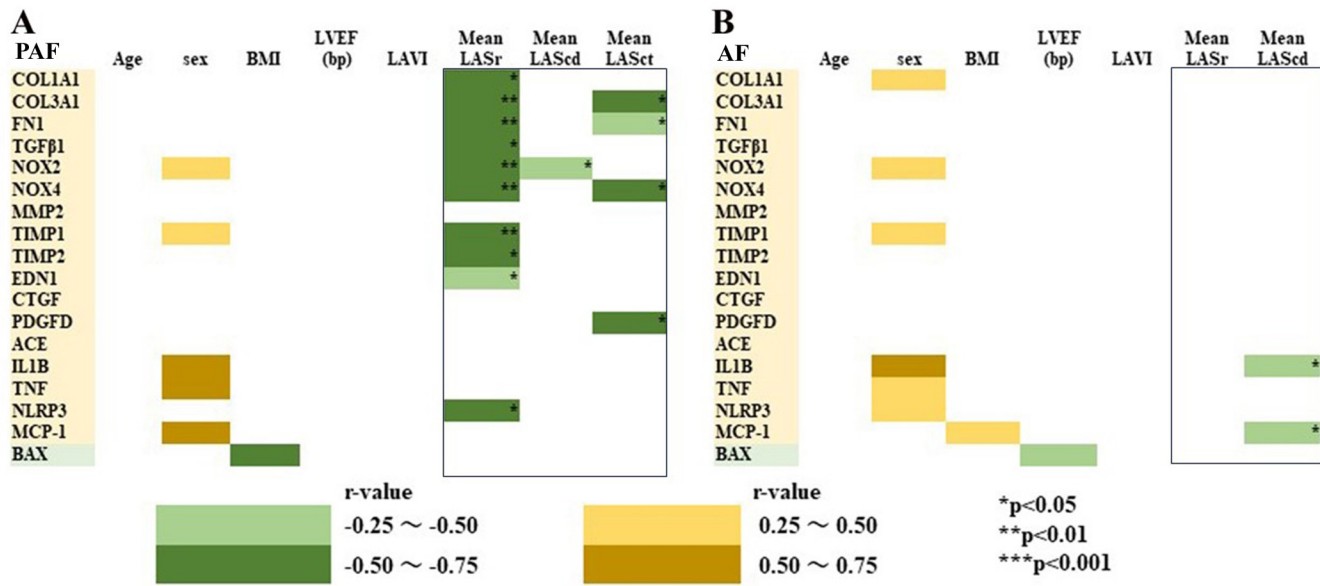

**Fig 7.** Correlation matrix between clinical data (age, sex, BMI), echocardiographic findings and fibrosis-related gene expression level in PAF (A) and AF (B) patients. The data were obtained from PAF patients (n = 19–20) and AF patients (n = 30–32) receiving LA resection. *R*–value and *P*–value in mLASr, mLAScd, and mLASct are shown. *P<0.05, **P<0.01, ***P<0.001. Significant negative relationships ($R = -0.25 \sim -0.5$, $R = -0.5 \sim -0.75$) are shown in green area, and positive relationships ($R = 0.25 \sim 0.5$, $R = 0.5 \sim 0.75$) are shown in orange area.

neither mLASr nor mLASct correlated with age, suggesting that the decrease in mLASr observed in this study was not due to aging. Several studies have shown an association between reduced LA reservoir function and PAF [37, 47, 48]. In this study, we did not differentiate patients with SR by the presence or absence of PAF in total patients. However, patients who underwent appendage resection had a history of PAF before surgery. Thus, low mLASr seems to be due in part to the presence of PAF. Further studies are needed to clarify whether LASr becomes a predictor of PAF in patients with SR undergoing cardiovascular surgery.

Several profibrotic signaling pathways have been implicated in atrial fibrosis and AF [2, 15, 16]. TGFβ1 plays a critical role in matrix remodeling and in enhancing collagen synthesis, and increased expression of TGFβ1 increases myocardial fibrosis [2, 15, 16]. A transgenic mouse model study also demonstrated that atrial fibrosis is a sufficient substrate for AF and that TGFβ1 plays an important role in the genesis of atrial fibrosis [17]. Thus, the role of TGFβ in atrial fibrosis and AF has been well established. In addition, the roles of oxidative stress in the mechanisms of the promotion of structural substrates for AF have been also well established [2, 15, 16]. Chang et al. [49] showed NOX2 upregulation in patients with AF, implicating a possible role of NOX2-derived oxidative stress in atrial remodeling in AF. In the present study, the univariate regression analysis among the fibrosis-related genes showed that TGFB1 was the strongest variable to predict the log (COL1A1) expression level in patients undergoing cardiovascular surgery. NOX4 and NOX2 were the next strongest genes to predict log (COL1A1) expression among genes examined in this study. Furthermore, multiple regression analysis showed that TGFB1 and NOX4 were independent determinants of COL1A1, even after adjusting for age, sex, and BMI. Thus, the present study provides the first evidence showing that TGFB1 and NOX are major determinants of atrial COL1A1 mRNA expression, even after adjusting for age, sex, and BMI in patients undergoing cardiovascular surgery.

A balance between MMPs and TIMPs ensures the maintenance of ECM homeostasis, but a lack of balance between MMPs and TIMPs translates into the extent of pathological

remodeling of myocardial ECM with intense fibrotic processes, which frequently take places during atrial remodeling [23]. The present study showed that MMPs and TIMPs appear to play a role in fibrotic remodeling in patients undergoing cardiovascular surgery. Inflammation is also an important contributing factor to AF development. The NLRP3 inflammasome is an inflammatory signaling complex that contributes to the formation of caspase-1 to produce the active forms of IL-1β, a pro-inflammatory cytokine [50]. The activity of the NLRP3 inflammasome is elevated in atrial myocytes from patients with paroxysmal and chronic AF [51]. Here, we demonstrated that atrial NLRP3 and IL1B mRNA levels, as well as levels of TNF and MCP-1, inflammatory cytokines, were associated with increased COL1A1 mRNA and fibrosis-related gene expression. Thus, it is very likely that inflammatory signals including the NLRP3 inflammasome lead to atrial fibrosis to contribute to AF in patients undergoing cardiovascular surgery, which is compatible with findings of previous papers [51]. Furthermore, several other profibrotic signaling pathways, such as ACE, PDGF, CTGF, ET-1, and apoptosis have also been reported to be implicated in myocardial fibrotic remodeling and AF [2, 15, 16]. Mayyas et al. [52] showed that the mRNA levels of collagen isoform 1, as well as collagen isoform 3, were each positively correlated with the atrial EDN1 mRNA level, and ET-1 mRNA levels were strongly associated with the expression of PDGFD and CTGF, suggesting that ET-1, PDGF and CTGF signaling are implicated as modulators of fibroblast proliferation. Similarly, in the present study, the COL1A1 expression level correlated with EDN1, PDGFD, and CTGF, suggesting that these signaling pathways are involved in atrial fibrosis. The present study also provided evidence for the involvement of ACE, the renin/angiotensin system and BAX, apoptosis, on atrial fibrosis and AF, which is compatible with findings of previous reports [24, 31]. However, the present study has investigated associations with the fibrosis-related gene expression levels, and therefore, further studies are required to clarify the detailed molecular mechanisms of each signaling pathway and the involvement of other pathways.

An increase in atrial pressure and mechanical stretch induces activation of the profibrotic signaling pathways, leading to fibroblast proliferation, collagen synthesis, and fibrosis, which contribute to the development of AF [2, 10, 11]. In fact, fibrosis-related gene expression showed a positive association with atrial ANP mRNA expression, reflecting increased mechanical stretching [53, 54], even in the PAF group. We also provided the first evidence that the degree of LA fibrosis, reflected by measurements of ECM (COL1A1) and fibrosis-related genes, correlated with mLASr. Multiple regression analysis showed that mLASr was a predictor of atrial COL1A1 expression, compatible with previous studies showing that LASr is correlated with LA fibrosis detected by DE-CMR or histology in patients with mitral valve disease [33, 34]. In contrast, LA volume has been used as an index for evaluating atrial structural remodeling [3, 4], but unlike mLASr, the LAVI did not correlate with fibrosis-related gene expression. LASr is the most widely used parameter, with more robust evidence than LAScd and LASct, and is independent of the presence of SR (ie, conversely LASct is absent in AF due to the lack of LA contraction) [55]. In the present study, among LAS, mLASr was a predictive of COL1A1 and the expression levels of other fibrosis-related genes. We also showed that mLASr was predictive of the atrial COL1A1 expression level, even in PAF patients. In those patients, mLASr was also negatively correlated with the fibrosis-related gene expression (TGFB1, NOX2, TIMP1, TIMP2, and NLRP3). Furthermore, PAF patients with low mLASr had higher expression levels of the fibrosis-related genes compared with those with high mLASr. Thus, it is very likely that low LASr in PAF patients reflects increased atrial fibrosis, and LASr is extremely useful as an early marker for predicting atrial fibrosis, and consequently AF, even before AF develops, in patients undergoing cardiovascular surgery.

LA reservoir function is influenced by LA relaxation/stiffness and LV contraction through the descent of the base during systole [1]. It is also altered secondary to increased LV filling

pressure and LVEDP with consequent mechanical stress on the LA leading to reduced reservoir function [8]. Reduced LASr (<18.0%) showed the highest diagnostic accuracy (sensitivity of 96%, specificity of 92%) in the prediction of elevated LVEDP (>12 mmHg) [8]. Several papers reported that low LASr is correlated with the pathophysiology of aortic stenosis (AS) [56]. As AS progresses, chronic increases in LV afterload and LV hypertrophy develop, which cause decreased LV compliance and an increase in LV filling pressure, resulting in LA dilatation, reduced LA compliance, and reduced LA reservoir function [56, 57]. Atrial remodeling in AS may serve as a substrate for AF [58]. The present study provides direct evidence that mLASr was an early marker for predicting LA fibrosis in patients undergoing cardiovascular surgery, including those with AS.

## Study limitations

Several limitations of the present study must be addressed. First, although measurement of LAS is a promising method for the evaluation of atrial mechanics, operator-dependence and resolution of imaging seem to be major limitations of this method [5], and it was difficult to obtain enough views for speckle tracking, probably due to physical differences, such as obesity. However, we could perform speckle-tracking in most (90%) participants. In addition, standard 4-chamber and 2-chamber views often maximize the long-axis of the left ventricle, resulting in artifactitious foreshortening of LA, which may overestimate LASr. To eliminate the foreshortened LA view, separate acquisition of ideal apical views that allows full visualization of the LA cavity is essential. However, even when extreme attention is paid to acquire non-foreshortened images at all tilt phases, it is virtually impossible to prevent minimal differences in views. Alternatively, the assessments of LA longitudinal strain using three-dimensional echocardiography may overcome it [59]. However, no consensus has been reached regarding the best methodology for LA strain measurements, and averaging views has never been demonstrated as superior from the diagnostic or prognostic point of view. Therefore, further studies are needed to clarify it. Thirdly, sample size was small, especially patients with PAF. However, it was sufficient to detect changes in the parameters of interest with statistical significance and thus increasing the sample size would not have changed the results of the study. Furthermore, all patients included in our study underwent cardiac surgery and had coronary artery disease or significant valve disease. The pathophysiologic mechanism of AF in patients with AF alone may be different. Therefore, our findings are not necessarily applicable to the general population of patients undergoing cardiovascular surgery or patients with AF alone. Most subjects had risk factors and were receiving medical treatment, and the use of drugs such as ACE-1/ARBs might have affected atrial fibrosis and AF [60]. However, the presence of cardiovascular risk factors and the use of drugs did not significantly affect the LAS value in the present study. Also, although the pathogenesis of AF is affected by many factors, we chose to examine the expression of genes selected. Therefore, the detailed molecular mechanisms and the downstream signaling pathways have not been investigated. Lastly, LA appendage tissue was obtained from PAF or AF patients due to ethical issues. New trials are needed to estimate the difference in these indicators between patients with PAF and healthy subjects. Thus, further detailed analyses of larger numbers of patients are required to clarify the roles of LAS in the general population of patients undergoing cardiovascular surgery.

## Clinical implications

Patients with cardiovascular disease develop alterations in LA geometry and function, and subsequently AF, known as LA remodeling with progressive atrial fibrosis. We showed that low

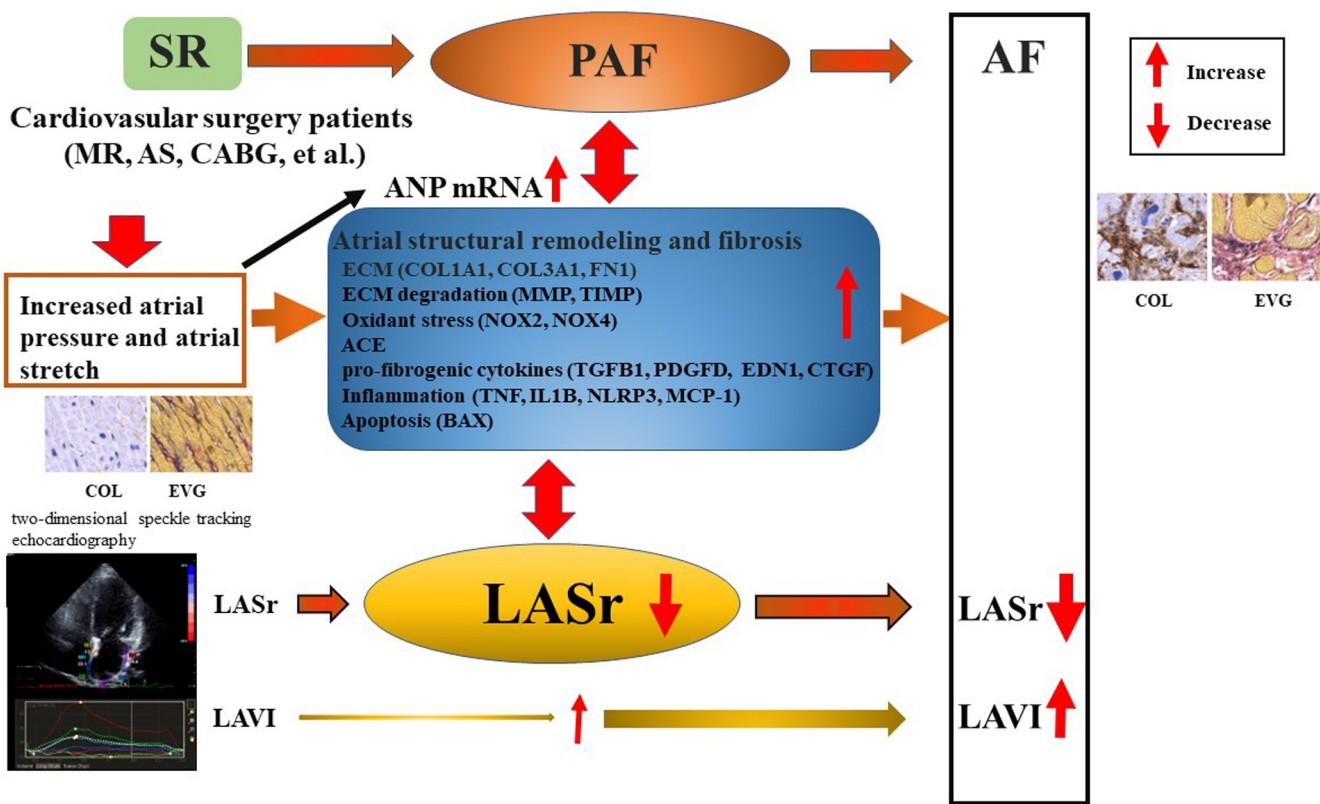

**Fig 8. Graphic abstract of the study.** SR, sinus rhythm; PAF, paroxysmal atrial fibrillation; AF, atrial fibrillation; ANP, atrial natriuretic peptide; LASr, left atrial reservoir strain; LAVI, left atrial volume index; COL collagen; EVG, Elastica van Gieson staining.

LASr reflects increased atrial fibrosis, and it is useful as a marker for predicting atrial fibrosis and AF in patients undergoing cardiovascular surgery.

## Conclusions

LASr correlated with atrial COL1A1 gene expression associated with fibrosis-related gene expression. Patients with low LASr exhibited increased atrial fibrosis-related gene expression, even those with PAF, highlighting the utility of LAS as a marker for LA fibrosis in cardiovascular surgery patients as illustrated in graphic abstract (Fig 8).

## Supporting information

**S1 Table. Primers used in this study.**
(DOCX)

**S2 Table. Relationships between left atrial strain (LAS) and clinical data (age, sex, BMI) and echocardiographic findings in SR patients including PAF and total patients.**
(DOCX)

**S3 Table. Correlations of risk factors and drug use with LAS parameters.**
(DOCX)

## Acknowledgments

We thank the cardiovascular surgeons and the staff members who collected the blood samples and atrial tissue samples, and Tokoi S for analyzing LAS in elderly control subjects.

## Author Contributions

**Conceptualization:** Toshiaki Nakajima.

**Data curation:** Akiko Haruyama, Kentaro Minami, Suguru Hirose.

**Formal analysis:** Taira Fukuda.

**Funding acquisition:** Toshiaki Nakajima.

**Investigation:** Akiko Haruyama, Hiroko Yazawa, Takafumi Nakajima, Takaaki Hasegawa, Yoshiyuki Kitagawa, Syotaro Obi, Gaku Oguri.

**Methodology:** Taira Fukuda.

**Project administration:** Toshiaki Nakajima, Shu Inami, Ikuko Shibasaki, Hirohisa Amano, Takuo Arikawa, Masashi Sakuma, Shichiro Abe.

**Supervision:** Hirotsugu Fukuda, Shigeru Toyoda.

**Writing – original draft:** Toshiaki Nakajima.

**Writing – review & editing:** Toshiaki Nakajima.

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
