## [Decision Letter · Decision Letter 0]

27 May 2024

PONE-D-24-10600Left atrial reservoir strain is a marker of atrial fibrotic remodeling in patients undergoing cardiovascular surgery: Analysis of gene expression.PLOS ONE

Dear Dr. Nakajima,

Thank you for submitting your manuscript to PLOS ONE. After careful consideration, we feel that it has merit but does not fully meet PLOS ONE’s publication criteria as it currently stands. Therefore, we invite you to submit a revised version of the manuscript that addresses the points raised during the review process.

We look forward to receiving your revised manuscript.

Kind regards,

Roberto Magalhães Saraiva, MD, PhD

Academic Editor

PLOS ONE

Journal Requirements:

"This study was supported in part by JSPS KAKENHI Grant Number 19H03981 (to T.N.), 22H03457 (T.N.) and the Vehicle Racing Commemorative Foundation (to T.N.)."

3. Please expand the acronym “JSPS” (as indicated in your financial disclosure) so that it states the name of your funders in full.

5. We notice that your supplementary tables are included in the manuscript file. Please remove them and upload them with the file type 'Supporting Information'. Please ensure that each Supporting Information file has a legend listed in the manuscript after the references list.

Additional Editor Comments:

This is a very interesting paper. However, there are issues that need to be addressed by the authors. Beware that beyond my remarks, all remarks from the reviewer should be equally addressed.

1) Include the “n” of the studied population for each Table and graph. Only part of the patients provided cardiac tissue for analysis, so the “n” of the studied population is not the same for each Table or Figure.

2) The so-called SR population is in fact composed by patients with paroxysmal atrial fibrillation. If patients with no previous episode of AF were included, differences could be greater between groups and some parameters that did not differ in the study could be different. Therefore, wherever applicable, the SR population should be renamed PAF population because in fact authors compared a population of patients with PAF with a population wit persistent/permanent AF.

3) What the etiology of the valvular disease? Rheumatic? Degenerative? It is important to understand the applicability of your results.

4) Please review your abbreviations so that they are defined the first time they appear in the text.

5) Correlations should be described as positive or negative and include the degree of the correlation (weak, moderate, strong).

6) Supplement Table 3. Include the r value.

Reviewers' comments:

Reviewer's Responses to Questions

**Comments to the Author**

1. Is the manuscript technically sound, and do the data support the conclusions?

Reviewer #1: Partly

2. Has the statistical analysis been performed appropriately and rigorously? 

Reviewer #1: Yes

3. Have the authors made all data underlying the findings in their manuscript fully available?

Reviewer #1: Yes

4. Is the manuscript presented in an intelligible fashion and written in standard English?

Reviewer #1: Yes

5. Review Comments to the Author

Reviewer #1: This is an interesting study - trying to correlate atrial function and left atrial structural and genetic remodeling and the role of atrial fibrillation. It is very difficult to realize the cause and effect (hen and egg sort of problem) and what is a marker of what. This goes for al levels of parameters included - LA volumes, function and gene expression.

some specific issues:

the study actually deals with 59 patients where genetic data were available. that should be emphasized and made clear to the reader.

LA active strain is only present in patients with SR, making this LASact only available for a small part of your patients.

Your patients were of course very sick with various situations that adversely affect myocardial function, and possibly very differently LA remodeling and function (just to compare mitral stenosis to ischemic cardiomyopathy). In any case the reported LASr range was very low in all patient compared to normal LASr. The cut off for prediction of AF was smaller than half the normal lowest LASr in the literature. This should be discussed, and is a major limitation of the manuscript.

Left ventricular function needs to be part of the analysis as it drive most of the passive LA function.

echocardiography: for valid speckle tracking of the LA - No foreshortening need to be assured. minimal volume should not disappear from images. Methods should provide information regarding avoidance of foreshortening. Intra-and inter observer variability regarding strain and volume assessments should be provided.

6. PLOS authors have the option to publish the peer review history of their article (what does this mean?). If published, this will include your full peer review and any attached files.

Reviewer #1: No

---

## [Author Response · Author response to Decision Letter 0]

5 Jun 2024

Reply to Reviewer 

We greatly appreciate your careful attention to our manuscript and especially your excellent suggestions for improving the clarity and correctness of the message. We have corrected the paper as per your suggestions, and consider the revised manuscript much improved.

#) This is a very interesting paper. However, there are issues that need to be addressed by the authors. Beware that beyond my remarks, all remarks from the reviewer should be equally addressed.

1) Include the “n” of the studied population for each Table and graph. Only part of the patients provided cardiac tissue for analysis, so the “n” of the studied population is not the same for each Table or Figure.

Response: Thank you very much for your comments. We added the “n” of the studied population. 

2) The so-called SR population is in fact composed by patients with paroxysmal atrial fibrillation. If patients with no previous episode of AF were included, differences could be greater between groups and some parameters that did not differ in the study could be different. Therefore, wherever applicable, the SR population should be renamed PAF population because in fact authors compared a population of patients with PAF with a population wit persistent/permanent AF.

Response: Thank you very much for your comments. SR patients who underwent LA dissection were renamed PAF patients. 

3) What the etiology of the valvular disease? Rheumatic? Degenerative? It is important to understand the applicability of your results.

Response: Thank you very much for your comments. We added the etiology of MR in Table 1, and mentioned it in results. 

 The etiology of MR (rheumatic, degenerative, prolapse, secondary, and others) is shown in Table 1.

4) Please review your abbreviations so that they are defined the first time they appear in the text.

Response: We checked about the abbreviations. 

5) Correlations should be described as positive or negative and include the degree of the correlation (weak, moderate, strong).

Response: Thank you very much for your comments. We described the correlations about the positive or negative value and the degree of the correlation (weak, moderate, strong).

The strength of correlation was set as follows: strong (R≧0.75), moderate (0.5≦R<0.75) and weak (0.5<R).

6) Supplement Table 3. Include the r value.

Response: Thank you very much for your comments. We added the r value.

Reply to Reviewer 

We greatly appreciate your careful attention to our manuscript and especially your excellent suggestions for improving the clarity and correctness of the message. We have corrected the paper as per your suggestions, and consider the revised manuscript much improved.

Reviewers' comments:

Reviewer's Responses to Questions

Comments to the Author

1. Is the manuscript technically sound, and do the data support the conclusions?

#) Response: Thank you very much for your suggestions. We added the control elderly data obtained from the subjects without PAF episodes and any organic heart diseases such as valvular diseases. We mentioned about it in discussions (unpublished results). Furthermore, in healthy individuals, LASr and LAScd have been reported to decrease with age. We mentioned about it in discussion as illustrated below (data not shown).

In our control data using elderly subjects without any organic heart diseases including valvular diseases (109 subjects, mean age 72 years, 57 males, unpublished data), mLASr, mLAScd, LASct and LAVI (bp) were 36.0% (33.6-39.3), 17.3% (15.0-21.6), 18.1% (15.2-21.0), and 25.6 ± 6.6, respectively. Thus, the patients studied in the present study had much lower LASr, LAScd, and LASct, and high LAVI (bp), compared with the control subjects. 

In healthy individuals, LASr and LAScd have been reported to decrease with age, [45, 46] which was confirmed in the control subjects (data not shown). However, in the present study of patients undergoing cardiovascular surgery, the mLAScd value decreased with age, but neither mLASr nor mLASct correlated with age, suggesting that the decrease in mLASr observed in this study was not due to aging.

 Control patients

Number 109

Male / Female, n (%) 57 (52) / 52 (48)

Age, y, median (range) 72.0 (68.0-77.0

BMI, kg/m2, median (IQR) or median (range) 22.6 ± 3.8

Atrial fibrillation, n (%) 0

Echocardiographic data, median (IQR) 

 LAD, mm 34 ± 5

 LVDd, mm 43 ± 5

 LVEF (bp), % 64.0 (62.0-66.0)

 LVMI, g/m2 67.5 (58.5-76.3)

 E/e’ 8.9 (7.4-10.6)

 LAVI (bp), ml/m2 25.6 ± 6.

 LASr 4CH, % 36.9 (34.1-40.7)

 LASr 2CH, % 35.7 (33.6-40.0)

 Mean LASr (mLASr), % 36.0 (33.6-39.3)

 LAScd 4CH, % 18.9 (14.9-24.9)

 LAScd 2CH, % 17.8 (14.1-22.8)

 Mean LAScd (mLAScd), % 17.3 (15.0-21.6)

 LASct 4CH, % 17.7 (14.4-20.5)

 LASct 2CH, % 17.4 (14.3-24.3)

 Mean LASct (mLASct), % 18.1 (15.2-21.0)

 control elderly subjects

 mLASr mLAScd mLASct

 r-value (p-value) r-value (p-value) r-value (p-value)

Age -0.552 (<0.001) *** -0.521 (<0.001) *** 0.093 (0.338)

Sex -0.277 (0.004) ** -0.074 (0.442) -0.153 (0.112)

BMI 0.083 (0.391) 0.030 (0.753) 0.084 (0.387)

#) Response We mentioned about sample size in discussions (Limitation). 

Second, sample size was small, especially patients with PAF. However, it was sufficient to detect changes in the parameters of interest with statistical significance and thus increasing the sample size would not have changed the results of the study. 

Reply to Reviewer 

We greatly appreciate your careful attention to our manuscript and especially your excellent suggestions for improving the clarity and correctness of the message. We have corrected the paper as per your suggestions, and consider the revised manuscript much improved.

Reviewer #1: This is an interesting study - trying to correlate atrial function and left atrial structural and genetic remodeling and the role of atrial fibrillation. It is very difficult to realize the cause and effect (hen and egg sort of problem) and what is a marker of what. This goes for all levels of parameters included - LA volumes, function and gene expression.

#) Response: Thank you very much for your comments. I absolutely agree with you. In this paper, we have mainly analyzed the relationships between LAS and atrial gene expression level.

Some specific issues:

the study actually deals with 59 patients where genetic data were available. that should be emphasized and made clear to the reader.

#) Response: Thank you very much for your comments. We described it in methods. Samples of the left atrial appendages (LAA) was obtained from 59 patients for whom genetic information was available. 

LA active strain is only present in patients with SR, making this LASact only available for a small part of your patients. Your patients were of course very sick with various situations that adversely affect myocardial function, and possibly very differently LA remodeling and function (just to compare mitral stenosis to ischemic cardiomyopathy). In any case the reported LASr range was very low in all patient compared to normal LASr. The cut off for prediction of AF was smaller than half the normal lowest LASr in the literature. This should be discussed, and is a major limitation of the manuscript.

Left ventricular function needs to be part of the analysis as it drive most of the passive LA function.

#) Response: Thank you very much for your comments. We agree with you. LA active strain is only present in patients with SR, making this LASact only available for a small part of our patients. We could not find significant correlation between LASct and gene expressions, in contrast with LASr. However, even in PAF patients, mLASr negatively correlated with extracellular matrix (COL1A1 (R=–0.539, P<0.05), COL3A1 (R=–0.642, P<0.01), and FN1 (R=–0.584, P<0.01) mRNA expression level. mLASct also negatively correlated with COL3A1 (R=–0.542, P<0.05), and FN1 (R=–0.488, P<0.05), but not with COL1A1. Thus, it is very likely that LASr is extremely useful as an early marker for predicting atrial fibrosis, and consequently AF, even before AF develops, in patients undergoing cardiovascular surgery.

#) As your suggestion, our reported LASr range was very low in all patients. We added the data obtained from control elderly subjects without AF and any cardiac diseases as shown below. 

In our control data using elderly subjects without any organic heart diseases including valvular diseases (109 subjects, mean age 72 years, 57 males, unpublished data), mLASr, mLAScd, LASct and LAVI (bp) were 36.0% (33.6-39.3), 17.3% (15.0-21.6), 18.1% (15.2-21.0), and 25.6 ± 6.6, respectively. Thus, the patients studied in the present study had much lower LASr, LAScd, and LASct, and high LAVI (bp), compared with the control subjects.

#) Response. As your suggestion, left ventricular function may affect LASr. However, in our study, LVEF did not significantly affect the gene expression of COL1A1 and the other fibrotic genes expression as shown in Fig 6. 

Echocardiography: for valid speckle tracking of the LA - No foreshortening need to be assured. minimal volume should not disappear from images. Methods should provide information regarding avoidance of foreshortening. Intra-and inter observer variability regarding strain and volume assessments should be provided.

#) Response: Thank you very much for your suggestions. We mentioned about it in limitations. We cited one paper.

In addition, standard 4-chamber and 2-chamber views often maximize the long-axis of the left ventricle, resulting in artifactitious foreshortening of LA, which may overestimate LASr. To eliminate the foreshortened LA view, separate acquisition of ideal apical views that allows full visualization of the LA cavity is essential. However, even when extreme attention is paid to acquire non-foreshortened images at all tilt phases, it is virtually impossible to prevent minimal differences in views. Alternatively, the assessments of LA longitudinal strain using three-dimensional echocardiography may overcome it.[59] However, no consensus has been reached regarding the best methodology for LA strain measurements, and averaging views has never been demonstrated as superior from the diagnostic or prognostic point of view. Therefore, further studies are needed to clarify it. 

59. Nabeshima Y, Kitano T, Takeuchi M. Reliability of left atrial strain reference values: A 3D echocardiographic study. PLoS One. 2021; 16(4): e0250089. https://doi.org/10.1371/journal.pone.0250089. PMID: 33852637.

#) Response: Thank you very much for your comments. We have checked about the inter-observer variability of LAS among two cardiac echocardiography specialists. We could obtain the good reliability of the measurement of LASr, LAScd. and LASct. The ICC (intraclass correlation coefficients) was 0.904 for mLASf, 0.938 for mLAScd, and 0.964 for mLASct, respectively. Furthermore, Cronbach’s coefficient alpha, another reliability coefficient, was 0.983 for mLASr (n=2). However, to clarify it, we need to increase n. Therefore, in this paper, we only mentioned about it in methods as follows. And, we cited several papers relating the reliability.

The analysis for LAS measurements using the intra- and inter-observer variability have been reported to show high reproducibility.[38-40] 

38. Genovese D, Singh A, Volpato V, Kruse E, Weinert L, Yamat M, et al. Load Dependency of Left Atrial Strain in Normal Subjects. J Am Soc Echocardiogr. 2018; 31(11): 1221-1228. https://doi.org/10.1016/j.echo.2018.07.016. PMID: 30205909.

39. Cameli M, Caputo M, Mondillo S, Ballo P, Palmerini E, Lisi M, et al. Feasibility and reference values of left atrial longitudinal strain imaging by two-dimensional speckle tracking. Cardiovasc Ultrasound. 2009; 7: 6. https://doi.org/10.1186/1476-7120-7-6. PMID: 19200402.

40. Ancona R, Comenale Pinto S, Caso P, Di Salvo G, Severino S, D'Andrea A, et al. Two-dimensional atrial systolic strain imaging predicts atrial fibrillation at 4-year follow-up in asymptomatic rheumatic mitral stenosis. J Am Soc Echocardiogr. 2013; 26(3): 270-277. https://doi.org/10.1016/j.echo.2012.11.016. PMID: 23261148.

---

## [Decision Letter · Decision Letter 1]

16 Jun 2024

Left atrial reservoir strain is a marker of atrial fibrotic remodeling in patients undergoing cardiovascular surgery: Analysis of gene expression.

PONE-D-24-10600R1

Dear Dr. Nakajima,

We’re pleased to inform you that your manuscript has been judged scientifically suitable for publication and will be formally accepted for publication once it meets all outstanding technical requirements.

Kind regards,

Roberto Magalhães Saraiva, MD, PhD

Academic Editor

PLOS ONE

Reviewers' comments:

Reviewer's Responses to Questions

**Comments to the Author**

1. If the authors have adequately addressed your comments raised in a previous round of review and you feel that this manuscript is now acceptable for publication, you may indicate that here to bypass the “Comments to the Author” section, enter your conflict of interest statement in the “Confidential to Editor” section, and submit your "Accept" recommendation.

Reviewer #1: All comments have been addressed

2. Is the manuscript technically sound, and do the data support the conclusions?

Reviewer #1: Yes

3. Has the statistical analysis been performed appropriately and rigorously? 

Reviewer #1: Yes

4. Have the authors made all data underlying the findings in their manuscript fully available?

Reviewer #1: (No Response)

5. Is the manuscript presented in an intelligible fashion and written in standard English?

Reviewer #1: Yes

6. Review Comments to the Author

Reviewer #1: (No Response)

7. PLOS authors have the option to publish the peer review history of their article (what does this mean?). If published, this will include your full peer review and any attached files.

Reviewer #1: No

---

## [Editor Report · Acceptance letter]

27 Jun 2024

PONE-D-24-10600R1 

PLOS ONE

Dear Dr. Nakajima, 

I'm pleased to inform you that your manuscript has been deemed suitable for publication in PLOS ONE. Congratulations! Your manuscript is now being handed over to our production team.

Kind regards, 

on behalf of

Dr. Roberto Magalhães Saraiva 

Academic Editor

PLOS ONE